

# Self-stigma and cognitive fusion in parents of children with autism spectrum disorder. The moderating role of self-compassion

Anna Pyszkowska, Krzysztof Rożnawski and Zuzanna Farny

University of Silesia, Katowice, Poland

## ABSTRACT

**Background:** Research shows that people with autism spectrum disorder and their families often experience social stigma. The internalization of social stigma can lead to the occurrence of self-stigma, understood as an internalized cognitive-affective self-directed and rigid process that results in individuals agreeing with stigmatizing opinions and applying them to themselves. Experiencing self-stigma can lead to a cognitive fusion with negative thoughts–especially those about oneself. Previous studies show that self-compassion reduces feelings of suffering, shame and self-stigma in a group of parents of children with autism spectrum disorder. The aim of this study was to test the relationship between self-stigma and cognitive fusion among parents of children with ASD. The moderating role of self-compassion as a protective factor was also verified.

**Methods:** The following questionnaires were used: Perceived Public Stigma Scale, Perceived Courtesy Stigma Scale, Self-Compassion Scale–Short Form, Cognitive Fusion Questionnaire, and Depression, Anxiety and Stress Scale. The study included 233 Polish parents of children with autism spectrum disorder (including 218 women).

**Results:** The results showed a positive correlation between fusion and both affiliate ($r = 0.31$, $p < 0.001$) and public stigma ($r = 0.33$, $p < 0.001$). Fusion and self-compassion were significant predictors of affiliate stigma. Self-compassion moderated the relationship between fusion and depression ($\beta = -0.11$, $p < 0.05$) and the relationship between fusion and stress ($\beta = -0.11$, $p < 0.05$). Cognitive fusion with negative beliefs about oneself can contribute to self-stigma. Defusion-oriented actions are an opportunity to distance oneself from emerging thoughts and eliminate their negative consequences. Self-compassion manifests itself in a compassionate and accepting attitude towards oneself and improves the individual's well-being. Actions taken to strengthen the indicated factors could contribute to a better quality of life of parents of children with ASD.

# INTRODUCTION

The third wave of cognitive-behavioral therapies (CBT) is characterized by themes such as metacognition, mindfulness, dialectics, cognitive fusion, acceptance of emotions

Corresponding author
Anna Pyszkowska,
anna.pyszkowska@us.edu.pl

(*Kahl, Winter & Schweiger, 2012*), or self-compassion (*Hayes, 2004*). In contrast to classic cognitive-behavioral therapy's focus on cognitive restructuring and emotion changing, the third wave concentrates primary on the psychological and behavioral processes associated with mental health and well-being, and their contexts (*Hofmann & Hayes, 2019*). Acceptance and Commitment Therapy (ACT) represents one of the third wave's approaches and its core goal is to develop psychological flexibility *via* acceptance, mindfulness and values-focused behavioral strategies (*Hayes, Strosahl & Wilson, 1999*). Research shows that ACT is effective in patients with anxiety disorders or depression (*Forman et al., 2007*), and its role is also significant when working with stigmatized groups of minorities (*Skinta et al., 2014*) such as parents of children with autism spectrum disorder (ASD, cf. *Lunsky, et al., 2017*).

Parents of children with ASD experience elevated levels of stress a lot of times due to different challenges they face during caregiving for their children (*Hayes & Watson, 2013*; *Chan & Lam, 2017*). Studies have compared the amount of stress parents of children with ASD experience to other groups, such as parents of children with typical development or those diagnosed with other disorders (*Hayes & Watson, 2013*): out of all those groups it has been proven that the levels of stress in families of children with ASD are the highest (*Hayes & Watson, 2013*). In addition, it can be observed that the risk of depression and anxiety within this group seems higher too (*Hamlyn-Wright, Draghi-Lorenz & Ellis, 2007*). Research shows that ACT is an effective approach in reducing depressive symptoms and enhancing well-being and psychological flexibility in parents of children with ASD (*Corti et al., 2018*; *Blackledge & Hayes, 2006*). *Harris (2012)* suggests that psychological flexibility, including defusion, acceptance and self-compassion may be considered as a protective factor while experiencing a gap between expectations and reality that cannot be changed.

One of the challenges parents of children with ASD face is the societal stigma directed towards their children and themselves (*Kinnear et al., 2016*). Due to certain features of ASD–poor social skills, inappropriate affective expressions, self-destructive acts, and language impairments–people with a diagnosis of ASD are likely to experience stigmatizing reactions from the general public (*Butler & Gillis, 2011*), such as insensitive comments, hostile stares, or even physical aggression (*Lovell & Wetherell, 2019*). Previous research highlights that stigmatized individuals may develop self-stigma, defined as an internalized cognitive-affective self-directed and rigid process that results in the individuals agreeing with stigmatizing opinions and applying them to themselves (*Smith-Palmer et al., 2020*). Self-stigma may comprise public stigma and affiliate stigma. Public stigma refers to stereotypical beliefs, prejudicial attitudes and discriminatory behaviors endorsed by a sizable group in society toward a discredited subgroup (*Corrigan & Watson, 2002*). Affiliate stigma refers to the extent of self-stigmatization among associates of the targeted minorities (*Mak & Cheung, 2010*), including parents of individuals with ASD (*Chan & Lam, 2017*).

Research shows that parents of children with ASD may experience being blamed by others as guilty for causing their children's disability (*Wong, Mak & Liao, 2016*), or seen as responsible for their children's behavior (*Myers, Mackintosh & Goin-Kochel, 2009*).

Therefore, they might be stigmatized alongside their children, which may result in affiliate stigma (*Chan & Lam, 2017*). This experience often leaves parents feeling humiliated, isolated, and judged (*Broady, Stoyles & Morse, 2017*), hence they may internalize the content of stigmatizing words and thoughts, and therefore social stigma can be transformed into self-stigma; when this happens, individuals seem to accept negative social judgments and incorporate them into their own value system (*Chan & Lam, 2018*). In consequence, the development of negative self-thoughts intensifies, hence one's self-esteem and self-efficacy may decrease, causing individuals to experience higher levels of stress and diminishing their well-being (*Corrigan, Rafacz & Rüsch, 2011*; *Lovell & Wetherell, 2019*). Based on these factors, self-stigma may make caregiving responsibilities more difficult for parents and play a negative role in their relationship with the child (*Chan & Lam, 2017*). Research suggests that acceptance and commitment therapy-based interventions may improve parents' well-being (*Corti et al., 2018*; *Blackledge & Hayes, 2006*), while internal resources such as defusion and self-compassion may act as potential protective factors against distress (*Marsh, Chan & MacBeth, 2018*; *Maisel et al., 2019*) and self-stigma (*Yadavaia & Hayes, 2021*).

Fusion and defusion are processes central to the psychological flexibility model that forms the basis of Acceptance and Commitment Therapy (*Hayes, Strosahl & Wilson, 1999*). Cognitive fusion refers to the tendency for behavior to be overly-regulated and influenced by cognition. The behavior and the experience of a "fused" person is dominated by thoughts, and as a consequence he/she becomes less sensitive to direct consequences in their environment (*Gillanders et al., 2014*). A person who experiences a rigidly fused relationship with their inner content will treat their thoughts with seriousness, as if the thoughts were a true narrative on their reality. This narrows down the ways in which the individual can think freely and take effective action in their life. The opposite—cognitive defusion—is an ability to see their experience as mere thoughts, not as a commanding reflection of a one true (and often very judgmental) reality.

The cognitive content of self-stigma usually consists of stereotypical beliefs aimed toward a discredited group (*Corrigan & Watson, 2002*). A person "fused" with stereotypical beliefs is going to act on them as though they were literally true. For example, a parent of a child with ASD "fused" with the thought "most people from my community would rather not be friends with families that have a member who has an ASD" most likely would abstain from participating in social events in the neighborhood and be hesitant about taking any action that could strengthen their relationship with a member of the community. Such a person is probably going to be insecure and tentative among others that know him or her to be a parent of a child with an ASD. Hence, experiencing stigma may heighten the probability of cognitive fusion, particularly for negative self-relevant thoughts. This may happen because experiencing a stigma increases the possibility of developing a distressing self-story. Research on internalized mental illness stigma showed that experiencing discrimination due to one's mental illness is associated with increased anticipated discrimination in the future, increased social stigma from others, and greater internalized stigma (*Quinn, Williams & Weisz, 2015*).

As proposed by *Neff (2003b)*, self-compassion shares three properties: self-kindness, common humanity and mindfulness. Respectively, they entail treating oneself in a friendly manner, seeing one's own difficulties as a part of the human condition, and being able to live one's own "stories" (self-narrative) gently, without overexaggerating. *Kirby, Tellegen & Steindl (2017)* conducted a meta-analysis on 21 randomized controlled trial studies examining the effects of compassion-based interventions. As a result, they found that self-compassion reduces human suffering, and specifically lowers levels of depression, anxiety and distress. There are also studies that underline self-compassion's role as a mediator between resilience in the face of adversity and mental well-being (*Trompetter, de Kleine & Bohlmeijer, 2017*; *Kotera, Van Laethem & Ohshima, 2020*). Self-compassion is related to higher life-satisfaction, lower levels of worry (*Mowlaie et al., 2017*) and posttraumatic growth (*Basharpoor, Mowlaie & Sarafrazi, 2020*), and it is negatively correlated with self-criticism (*Neff & McGeheea, 2010*). *Luoma & Platt (2015)* indicate that ACT processes are to some extent practically self-compassionate. They further imply that shame is the core emotion in the experience of stigma (*Luoma et al., 2012*). Shame has a socially distancing and isolating effect (*Dickerson, Gruenewald & Kemeny, 2004*; *Dorahy, 2010*), as a person "fused" with shame-related cognitions will narrow down their behavior to such that will reflect the "fused" narrative and provide relief from unwanted but self-generated feelings. Compassion has a tendency to coexist with a different set of emotions such as sympathetic joy, warmth, interest and pride (*Goetz, Keltner & Simon-Thomas, 2010*) and endorse different behaviors. To be self-compassionate means to relate kindly to one's own experiences and to be caring towards oneself (*Luoma & Platt, 2015*). *Luoma & Platt (2015)* argue that self-compassion is an implicit ACT process that has an impact on the effectiveness of ACT interventions for shame and stigma. ACT proved effective in reducing shame related to substance addiction (*Luoma et al., 2008*), self-stigma related to obesity (*Lillis et al., 2009*), sexuality (*Yadavaia & Hayes, 2021*), psychological disorders (*Masuda et al., 2007*) and HIV when combined with Compassion Focused Therapy (*Skinta et al., 2014*). *Vowles et al. (2014)* found that self-compassion was the strongest mediator in justifying the positive effect of ACT among patients with chronic pain.

While having certain emotions is not typically associated with either the presence or the absence of fusion, chronic shame, anxiety, and depression can be seen as a reflection of an underlying narrow, rigid and constricting "fused" relationship with own inner content (*Valvano et al., 2016*). Seen this way, a self-compassionate perspective on oneself and others should undermine this schematic, chronic pattern of thinking and feeling. Also, it is worth noting that self-compassion was significantly and negatively correlated with cognitive fusion in patients with chronic pain cognitive fusion mediated the relationship between the chronic pain and depression symptoms, and self-compassion at all levels moderated that correlation; (*Carvalho et al., 2018*), in individuals with childhood abuse experience (*Basharpoor, Mowlaie & Sarafrazi, 2020*) and in cancer patients (*Gillanders et al., 2015*). Self-compassion was found to be the moderator between affiliate stigma and distress in parents of children with autism spectrum disorder (*Wong, Mak & Liao, 2016*). The predictive effect of self-stigma on psychological variables such as depression, somatic

symptoms, health status or quality of life was reduced by self-compassion by approximately one-third in overweight and obesity (*Hilbert et al., 2015*).

The current study design was similar to the one presented by *Chan & Lam (2017)*. The aim of the current study was to determine relationships between self-stigma and cognitive fusion and their effect on affective symptoms in parents of children with ASD. The study focused on whether self-stigma (affiliate stigma and public stigma) and cognitive fusion predicted depression, anxiety and stress in parents of children with ASD. Also, it was decided to add self-compassion to the model as it was reported to decrease the effect of self-stigma-related attitudes in minority groups (*Wong, Mak & Liao, 2016*) and to be highly associated with mindfulness (*Neff, 2003a*) also reported as a protective factor against self-stigma (*Chan & Lam, 2018*). It was hypothesized that affiliate stigma and public stigma would be linked with higher levels of depression, anxiety and stress, and increase with higher levels of cognitive fusion. Moreover, it was hypothesized that parents with a more self-compassionate attitude would present weaker links between self-stigma, fusion and affective symptoms.

## MATERIALS AND METHODS

This study was approved by the Ethics Committee of the University of Silesia in Katowice (approval no.: KEUS.92/02.2021). All participants provided written informed consent prior to enrolment in the study.

Nonprobability sampling was used. The inclusion criterion was being a parent of a child with a diagnosis of autism spectrum disorder (F84.0). Participants were recruited in schools, kindergartens and ASD-dedicated foundations in Poland *via* Internet. A total of 233 parents of children with ASD participated in the present study using an on-line survey. The majority of parents were female (93%), most of them had higher education (57%) and were married (71%). Most participants worked part-time (46%) or full-time (43%) jobs. The majority of children were male (84%) and had no intellectual disability (52%). Sociodemographic data is presented in Table 1. Despite the noticeable disproportion in terms of gender in favor of females, it was decided to use data from both the mothers and the fathers. This was due to the fact that a significant part of the research conducted so far has also faced the problem related to the small amount of data regarding experiences of fathers of children with ASD, so it was decided to take into account at least a small part of it, with the proviso that the results obtained could be generalized to reflect the experiences of all men.

To measure the variables, sociodemographic metrics and five questionnaires were used. Polish translation of two questionnaires was carried out for the purpose of the study: Perceived Public Stigma Scale (*Chan & Lam, 2017*) and Perceived Courtesy Stigma Scale (*Chan & Lam, 2017*) using the back translation method. No pilot testing was conducted on the normative sample due to the specific content of both questionnaires aimed at parents of children with autism spectrum disorder (*Chan & Lam, 2017*). The translators were granted approval by the authors of the original questionnaires. Participants who decided to be involved in the study filled by themselves the following tools:

| Table 1 Sociodemographic data. | | |
|---|---|---|
| | N = 233 | % |
| **Age of parent** | | |
| Range | 21–67 | |
| Mean | 39.18 | |
| Standard deviation | 7.79 | |
| **Gender of parent** | | |
| Male | 15 | 6.44 |
| Female | 218 | 93.56 |
| **Level of education** | | |
| Primary education | 7 | 3.00 |
| Secondary education | 93 | 39.91 |
| Higher education | 133 | 57.08 |
| **Professional status** | | |
| Full-time job | 100 | 42.92 |
| Part-time job | 108 | 46.35 |
| Unemployed | 25 | 10.73 |
| **Marital status** | | |
| Single | 7 | 3.00 |
| Married | 166 | 71.24 |
| Informal relationship | 29 | 12.45 |
| Divorced | 29 | 12.45 |
| Widow/-er | 2 | .86 |
| **Age of child** | | |
| Range | 2–37 | |
| Mean | 10.15 | |
| Standard deviation | 5.44 | |
| **Gender of child** | | |
| Male | 196 | 84.12 |
| Female | 37 | 15.88 |
| **Level of child's intellectual disability** | | |
| No intellectual disability | 122 | 52.36 |
| Mild | 53 | 22.75 |
| Moderate | 36 | 15.45 |
| Severe | 22 | 9.44 |

Public stigma. Perceived public stigma was measured using the Perceived Public Stigma Scale (PPSS; *Chan & Lam, 2017*, Polish translation for the purpose of this study by A Pyszkowska & K Rożnawski, 2020, unpublished). The Scale contained eight items adapted from Green's (2001) study. A sample item was "Most people feel that having an ASD is a sign of personal failure". Participants rated each item on a six-point Likert scale ranging from 0 (*strongly disagree*) to 5 (*strongly agree*). In the present study, Cronbach's alpha was $\alpha = 0.88$.

Affiliate stigma. Affiliate stigma was measured using the Perceived Courtesy Stigma Scale (PCSS; *Chan & Lam, 2017*, Polish translation for the purpose of this study by A Pyszkowska & K Rożnawski, 2020, unpublished), which contained seven items adapted from the Devaluation of Consumer Families Scale by *Struening et al. (2001)*. A sample item was "Most people do blame parents for the ASD of their children". Participants rated each item on a four-point Likert scale ranging from 0 (*strongly disagree*) to 3 (*strongly agree*). In the present study, Cronbach's alpha was α = 0.86.

Self-compassion. Self-compassion was measured using the Self-Compassion Scale–Short Form (SCS-SF; *Raes et al., 2011*; Polish translation by Kocur D., 2016, unpublished), which contained 12 items adapted from the Self-Compassion Scale (SCS; *Neff, 2003a*). A sample item was "When I'm going through a very hard time, I give myself the caring and tenderness I need". Participants rated each item on a five-point Likert scale ranging from 1 (*almost never*) to 5 (*almost always*). In the present study, Cronbach's alpha was α = 0.83.

Cognitive fusion. Cognitive fusion was measured using the Cognitive Fusion Questionnaire (*Gillanders et al., 2014*; Polish translation: *Baran, Hyla & Kleszcz, 2019*) which contained seven items (*e.g.*, "I struggle with my thoughts"). Participants rated each item on a seven-point Likert scale ranging from 1 (*never true*) to 7 (*always true*). In the present study, Cronbach's alpha was α = 0.93.

Depression, anxiety and stress. Depression, anxiety and stress symptoms were measured through the Depression, Anxiety and Stress Scale (DASS-21, *Lovibond & Lovibond, 1995*, Polish translation by *Lewicka et al., 2013*) which contained twenty-one items. A sample item was "I felt life was meaningless". Participants rated each item on a four-point Likert scale ranging from 0 (*rarely or never*) to 3 (*most of the time*). In the present study, Cronbach's alpha was as following: α for depression = 0.93, α for anxiety = 0.94, α for stress = 0.92.

# RESULTS

All calculations were performed with the use of Jamovi 1.6.23.

Demographic variables were analyzed through descriptive statistics. Bivariate correlation analyses were performed to examine relationships between independent variables (affiliate stigma, public stigma, cognitive fusion), the moderator variable (self-compassion), and dependent variables (depression, anxiety, stress). First, a hierarchical multiple regression model was used in order to determine predictors (affiliate stigma, public stigma, fusion) of depression, anxiety and stress while controlling for parent's professional status, parent's level of education, and the child's level of intellectual disability. Then, six hierarchical multiple regression models were conducted including an interaction term between two predictors: self-stigma and fusion, and a moderator (self-compassion). The independent and moderator variables were standardized before the interaction term was computed to reduce multicollinearity. In step 1, standardized covariates were entered (parent's professional status, parent's level of education and child's level of intellectual impairment). In step 2, standardized independent and moderator variables were added, and finally, in step 3, two separate interactions between predictors
| | M | SD | 1. | 2. | 3. | 4. | 5. | 6. | 7. |
|---|---|---|---|---|---|---|---|---|---|
| **Table 2 Descriptive statistics and Pearson's r correlations.** | | | | | | | | | |
| 1. Affiliate stigma | 18.50 | 8.49 | – | | | | | | |
| 2. Public stigma | 23.77 | 6.68 | 0.47*** | – | | | | | |
| 3. Self-compassion | 33.28 | 9.06 | −0.24*** | −0.32*** | – | | | | |
| 4. Cognitive fusion | 31.37 | 11.11 | 0.31*** | 0.33*** | −0.64*** | – | | | |
| 5. Depression | 16.65 | 10.55 | 0.33*** | 0.36*** | −0.54*** | 0.66*** | | | |
| 6. Anxiety | 18.96 | 11.14 | 0.35*** | 0.38*** | −0.59*** | 0.64*** | 0.92*** | | |
| 7. Stress | 14.02 | 10.86 | 0.35*** | 0.37*** | −0.52*** | 0.65*** | 0.95*** | 0.94*** | – |

**Note:**
*** $p < 0.001$.

(affiliate or public stigma and fusion) and moderator (self-compassion) were included. For statistically significant interactions, simple slope tests were conducted to examine whether the effects of the independent variables on the dependent variables were significant for high (+1 SD), medium (1 SD) and low (−1 SD) levels of the moderator.

ANOVA test showed no significant differences between parents of children with various levels of ASD-related intellectual disability except for affiliate stigma (F = 2.60, $p$ = 0.05). Bonferroni-corrected contrasts revealed differences between parents of children with moderate and severe intellectual disability (t = −2.55, $p$ = 0.05). Therefore, it was decided to conduct all analyses without grouping. Descriptive statistics and Pearson's r correlations between variables studied are summarized in Table 2.

Above-average rates of depression (M = 16.65), anxiety (M = 18.96) and stress (M = 14.02) were recognized. Both affiliate (r = 0.31, $p$ < 0.001) and public stigma (r = 0.33, $p$ < 0.001) showed significant positive associations with fusion, as well as with depression ($r_{affiliate}$ = 0.33, $p$ < 0.001, $r_{public}$ = 0.36, $p$ < 0.001), anxiety ($r_{affiliate}$ = 0.35, $p$ < 0.001, $r_{public}$ = 0.38, $p$ < 0.001) and stress ($r_{affiliate}$ = 0.35, $p$ < 0.001; $r_{public}$ = 0.37, $p$ < 0.001). Self-compassion showed negative correlations with both affiliate (r = −0.24, $p$ < 0.001) and public stigma (r = −0.32, $p$ < 0.001). Fusion was positively correlated with all negative symptoms (r range = 0.64–0.66, $p$ < 0.001) while self-compassion showed negative associations (r range = −0.52 to −0.59, $p$ < 0.001).

Tables 3 and 4 present hierarchical regression models.

The hierarchical regression model revealed that in step 1, background factors predicted 2% (F = 2.837) of depression, with significance of the parental education level (β = 0.20, $p$ < 0.01). An addition of main effects explained a further 48% (F = 39.077) of variance (affiliate stigma β = 0.16, $p$ < 0.01; self-compassion β = −0.17, $p$ < 0.01; fusion β = 0.50, $p$ < 0.001). Interaction effects explained an additional 1%, (F = 30.530) with the significance of the moderating effect on fusion (β = −0.10, $p$ < 0.05). Simple slope tests showed (see Fig. 1) that a high level of self-compassion was associated with a smaller (β = 4.85, SE = 0.59, $p$ < 0.001) effect of fusion on depressive symptoms when compared to low rates of self-compassion (β = 6.90, SE = 0.53, $p$ < 0.001). In the case of anxiety, background factors explained 4% (F = 3.044) with the parental education level being the only significant factor (β = 0.19, $p$ < 0.05), while main effects constituted a further 47%

**Table 3 Hierarchical regression model, affiliate stigma.**

| | Depression | | | Anxiety | | | Stress | | |
|---|---|---|---|---|---|---|---|---|---|
| | Step 1 | Step 2 | Step 3 | Step 1 | Step 2 | Step 3 | Step 1 | Step 2 | Step 3 |
| Background factors, β | | | | | | | | | |
| Professional status | −0.05 | −0.02 | −0.03 | −0.06 | −0.03 | −0.03 | −0.06 | −0.02 | −0.03 |
| Parental education level | 0.20** | 0.20*** | 0.20*** | 0.19* | 0.19*** | 0.19*** | 0.16* | 0.16** | 0.16*** |
| Level of child's intellectual disability | 0.04 | 0.02 | 0.03 | 0.10 | 0.08 | 0.09 | 0.10 | 0.08 | 0.09 |
| Main effects, β | | | | | | | | | |
| Affiliate stigma | | 0.16** | 0.16*** | | 0.18*** | 0.19*** | | 0.18*** | 0.19*** |
| Self-compassion | | −0.17** | −0.18** | | −0.27*** | −0.28*** | | −0.14* | −0.15** |
| Cognitive fusion | | 0.50*** | 0.53*** | | 0.41*** | 0.43*** | | 0.50*** | 0.53*** |
| Interaction effects, β | | | | | | | | | |
| Affiliate stigma x self-compassion | | | −0.04 | | | −0.06 | | | −0.10* |
| Cognitive fusion x self-compassion | | | −0.10* | | | −0.04 | | | −0.07 |
| ΔR | 0.02 | 0.48 | 0.01 | 0.04 | 0.47 | 0.00 | 0.01 | 0.46 | 0.02 |

Notes:
  * $p < 0.05$.
  ** $p < 0.01$.
  *** $p < 0.001$.

**Table 4 Hierarchical regression model, public stigma.**

| | Depression | | | Anxiety | | | Stress | | |
|---|---|---|---|---|---|---|---|---|---|
| | Step 1 | Step 2 | Step 3 | Step 1 | Step 2 | Step 3 | Step 1 | Step 2 | Step 3 |
| Background factors, β | | | | | | | | | |
| Professional status | −0.05 | −0.04 | −0.05 | −0.06 | −0.05 | −0.05 | −0.06 | −0.04 | −0.05 |
| Parental education level | 0.20** | 0.17*** | 0.18*** | 0.19* | 0.16*** | 0.16*** | 0.16* | 0.13** | 0.13** |
| Level of child's intellectual disability | 0.04 | 0.02 | 0.01 | 0.10 | 0.07 | 0.07 | 0.10 | 0.07 | 0.07 |
| Main effects, β | | | | | | | | | |
| Public stigma | | 0.13** | 0.14** | | 0.15** | 0.16** | | 0.15** | 0.16*** |
| Self-compassion | | −0.16** | −0.17** | | −0.26*** | −0.27*** | | −0.13* | −0.14* |
| Cognitive fusion | | 0.54*** | 0.54*** | | 0.43*** | 0.44*** | | 0.51*** | 0.54*** |
| Interaction effects, β | | | | | | | | | |
| Public stigma x self-compassion | | | −0.03 | | | −0.04 | | | −0.04 |
| Cognitive fusion x self-compassion | | | −0.11* | | | −0.05 | | | −0.11* |
| ΔR | 0.02 | 0.47 | 0.01 | 0.04 | 0.47 | 0.00 | 0.01 | 0.45 | 0.02 |

Notes:
  * $p < 0.05$.
  ** $p < 0.01$.
  *** $p < 0.001$.

(F = 42.109; affiliate stigma β = 0.18, $p < 0.001$; self-compassion β = −0.27, $p < 0.001$; fusion β = 0.41, $p < 0.001$). An addition of interaction effects did not enhance the variance (F = 32.112). Background factors predicted 1% of stress (F = 2.117; parental education level β = 0.16, $p < 0.05$), an addition of main effects constituted another 46% (F = 35.848; affiliate stigma β = 0.18, $p < 0.001$; self-compassion β = −0.14, $p < 0.05$; fusion β = 0.50, $p < 0.001$).

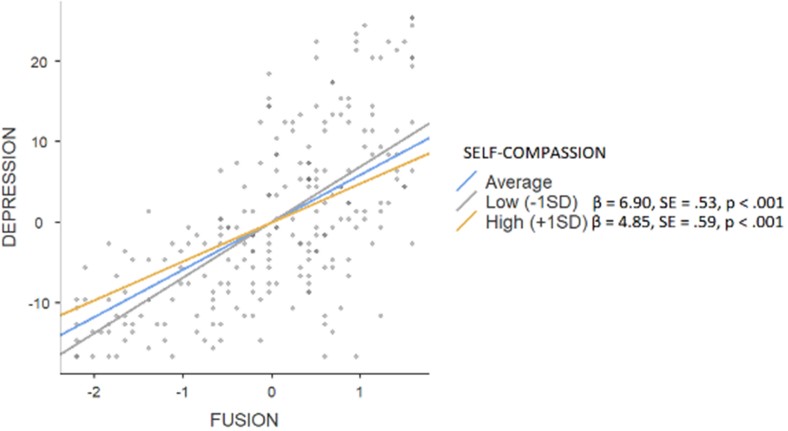

**Figure 1 Simple slope test, moderating effect of self-compassion between fusion and depression.**

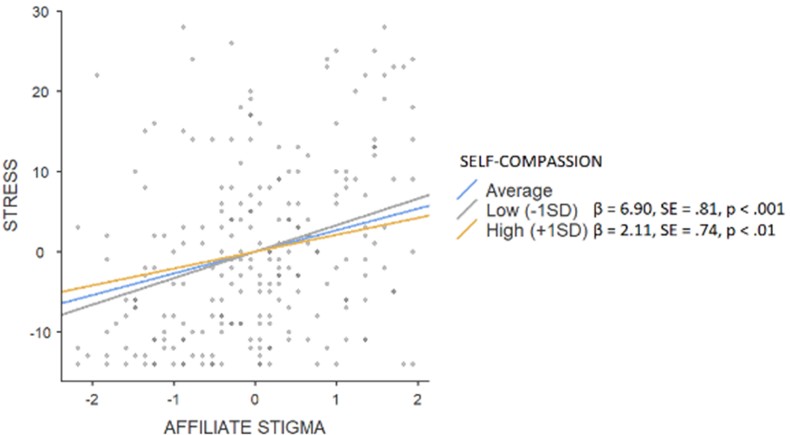

**Figure 2 Simple slope test, moderating effect of self-compassion between affiliate stigma and stress.**

Interaction effects explained an additional 2%, (F = 28.469) with the significance of the moderating effect on affiliate stigma (β = −0.10, $p < 0.05$). Simple slope tests showed (see Fig. 2) that a high level of self-compassion was associated with a lower effect of affiliate stigma on stress (β = 2.11, SE = 0.74, $p < 0.01$) when compared to low rates of self-compassion (β = 6.90, SE = 0.81, $p < 0.001$).

The hierarchical regression model revealed that in step 1, background factors predicted 2% of depression (F = 2.837), with significance of the parental education level (β = 0.20, $p < 0.01$). An addition of main effects explained a further 47% of variance (F = 37.995; public stigma β = 0.13, $p < 0.01$; self-compassion β = −0.16, $p < 0.01$; fusion β = 0.54, $p < 0.001$). Interaction effects explained an additional 1%, (F = 29.752) with the significance of the moderating effect on fusion (β = −0.11, $p < 0.05$). Simple slope tests showed (see Fig. 3) that high level of self-compassion was associated with a smaller (β = 4.85, SE = 0.59, $p < 0.001$) effect of fusion on depressive symptoms when compared to low rates of self-compassion (β = 6.90, SE = 0.81, $p < 0.001$). In the case of anxiety,

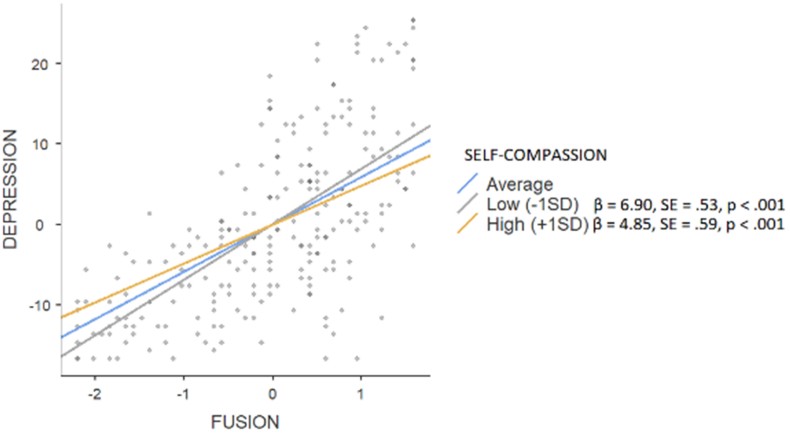

**Figure 3 Simple slope test, moderating effect of self-compassion between fusion and depression.**

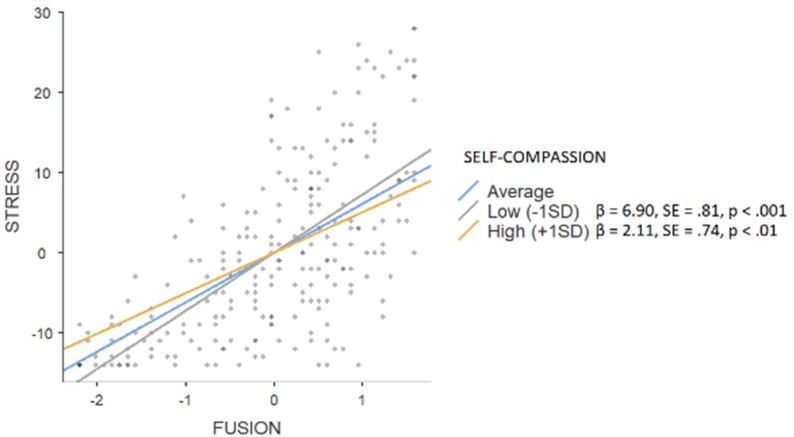

**Figure 4 Simple slope test, moderating effect of self-compassion between fusion and stress.**

background factors explained 4% (F = 3.044) with the parental education level being the only significant factor ($\beta$ = 0.19, $p < 0.05$), while main effects constituted a further 47% (F = 40.594; public stigma $\beta$ = 0.15, $p < 0.01$; self-compassion $\beta$ = −0.26, $p < 0.001$; fusion $\beta$ = 0.43, $p < 0.001$). An addition of interaction effects did not enhance the variance (F = 30.873). Background factors predicted 1% of stress (F = 2.117; parental education level $\beta$ = 0.16, $p < 0.05$), an addition of main effects constituted another 45% (F = 34.605; public stigma $\beta$ = 0.15, $p < 0.01$; self-compassion $\beta$ = −0.13, $p < 0.05$; fusion $\beta$ = 0.51, $p < 0.001$). Interaction effects explained an additional 2%, (F = 27.42) with the significance of the moderating effect on fusion ($\beta$ = −0.11, $p < 0.05$). Simple slope tests showed (see Fig. 4) that a high level of self-compassion was associated with a lower effect of fusion on stress ($\beta$ = 2.11, SE = 0.74, $p < 0.01$) when compared to low rates of self-compassion ($\beta$ = 6.90, SE = 0.81, $p < 0.001$).

## DISCUSSION

The aim of this study was to verify associations between self-stigma, cognitive fusion and self-compassion in parents of children with autism spectrum disorder. It was confirmed that both affiliate (r = 0.31, $p < 0.001$) and public stigma (r = 0.33, $p < 0.001$) were moderately positively associated with cognitive fusion. As the roots of self-stigma are defined as rigid and conceptualized assumptions about oneself derived from opinions and beliefs regarding the object of stigma and the social image of the stigmatized person (*Smith-Palmer et al., 2020*), one might argue that self-stigma would be interpreted as a fusion with these assumptions, *e.g.*, "since I am a bearer of a stigma, hence I am inherently defined by it". As cognitive fusion is strongly associated with rigidity and narrow perspective (*Hayes, Strosahl & Wilson, 1999*), it could be hypothesized that it would maintain or increase one's self-stigma narrative about oneself. On the contrary, defusion would act as a potential protective factor aimed at widening one's perspective and decrease the impact of self-stigma assumptions on the individual's mental health and well-being. On the other hand, the study showed an apparent negative link between self-compassion and fusion (r = −0.64, $p < 0.001$) and moderate correlations with affiliate (r = −0.24, $p < 0.001$) and public stigma (r = −0.32, $p < 0.001$). It could be considered to be in line with previous research regarding compassionate attitude towards oneself as flexible, non-judgmental (*Neff & Tirch, 2013*) and negatively associated with fusion (*Carvalho et al., 2018*; *Basharpoor, Mowlaie & Sarafrazi, 2020*) on the one hand, and negatively related to self-stigma (*Wong, Mak & Liao, 2016*) on the other hand.

Both models tested revealed that self-stigma (public and affiliate), self-compassion and fusion acted as significant predictors of depression, anxiety and stress among parents of children with autism spectrum disorder. Background factors and main effects accounted for 50% of depression (F = 39.077), 51% of anxiety (F = 42.109), and 47% of stress (F = 35.848) when affiliate stigma was used as a predictor, and 49% of depression (F = 37.995), 51% of anxiety (F = 40.594), and 46% of stress (F = 34.605) when public stigma was used as a predictor. These results may suggest that it is not only public or affiliate stigma that affects psychological suffering in parents of children with ASD, but mostly their psychological resources, as fusion and self-compassion accounted for the highest rates of the estimates. As *Hayes & Hofmann (2017)* suggest, psychological resources such as flexibility are somewhat independent from environmental factors, hence it is the function, not content, of thoughts or attitudes that is of highest importance for one's relation to reality and experiences such as social comments or stigmatization. Additionally, the parent's level of education was also a significant predictor, which is in line with previous research (*Chan & Lam, 2017*), although this variable presented far smaller estimates when compared to psychological functions.

Self-compassion acted as a significant moderator between affiliate stigma and depression as well as fusion and depression or stress. In all cases, higher levels of self-compassion were associated with smaller effects of fusion on psychological suffering (depressive or stress symptoms) or affiliate stigma on stress. Of note, self-compassion did not moderate relationships between self-stigma and anxiety perceived by participants

of this study. It can be hypothesized that anxiety accompanies families of people with autism spectrum disorder in various contexts, irrelevant of the stigmatizing opinions of others or the parents themselves (*e.g.*, fear of poor social support; dependence; finances, cf. *Myers, Mackintosh & Goin-Kochel, 2009*). The obtained results are in line with the reports by *Wong, Mak & Liao (2016)*, as well as with the research by *Chan & Lam (2017)* using mindfulness as a moderator between self-stigma and depression and anxiety symptoms. It could be hypothesized that a compassionate attitude towards oneself may foster awareness of inner experience without suppression and over-identification (*Neff, 2003a*): thus, the stigmatized person can take a more level-headed view of the current situation and accept all aspects of their identity as they are. As a result, self-stigmatizing thoughts may no longer be the cause of psychological suffering, *e.g.*, depressive or anxiety symptoms–although the aim of developing self-compassion or defusion is not to erase negative thoughts (*Hayes, Strosahl & Wilson, 1999*), the decreasing of psychopathology can be considered as a side effect of their development (*Chin & Hayes, 2017*).

Research shows that individuals with high self-compassion are capable of distinguishing between their sensory experience and mental experience of stigmatization, hence they would probably internalize the stigma (*Wong et al., 2018*), as well as may less likely suppress undesired thoughts and feelings (*Neff, 2003a*), acting more cognitively flexible (*Martin, Staggers & Anderson, 2011*). Both suppressing one's thoughts and feelings (experiential avoidance) and lack of cognitive flexibility are elements of cognitive fusion. Thus, it can be hypothesized that self-compassion may influence cognition in a way that protects the individual from cognitive fusion.

Of note, the level of the child's intellectual impairment did not differentiate most results, except for affiliate stigma, nor did it act as a significant predictor of the parents' symptoms. Affiliate stigma was higher in parents of children with moderate or severe intellectual disability, which can lead to a hypothesis regarding the social reception of people with intellectual impairments and their families (*Beighton & Wills, 2017*; *Mitter, Ali & Scior, 2019*). This issue could be considered in line with reports of higher social exclusion among people with severe autism spectrum disorder symptoms when compared to those with less strong symptoms (*Mitter, Ali & Scior, 2019*). Once more, it can be hypothesized that it is not the severity of the child's symptoms that is critical in the case of the parents' experience of care; however, due to the insufficient amount of data in this area in the current study, these assumptions are of purely speculative nature and require further research.

## CONCLUSIONS

The current study showed that self-compassion may act as a significant moderator between affiliate stigma, fusion and depression or stress. Parents of children with autism spectrum disorder characterized by higher intensity of self-compassion were less likely to experience negative symptoms due to self-stigmatizing thoughts when compared to those with low self-compassion. Additionally, the results obtained suggest that cognitive fusion is moderately related to self-stigma (both affiliate and public) which

may suggest that a rigid and narrow perspective about oneself is a psychological mechanism that underlies self-stigma. Hence, interventions focused on the development of defusion and self-compassion, not self-stigma *per se*, might be considered as imperative in reducing negative effects of self-stigma and distress.

Despite its strengths such as an adequate sample size (N = 233), the present study has certain limitations. It must be highlighted that the majority of the study participants were female, hence the results obtained must be considered more as an experience of mothers than of fathers. Future research should focus on developing knowledge regarding the experiences of fathers of children with ASD, as it can be hypothesized that it varies from the ones reported by the mothers, as research shows significant gender differences *e.g.*, in terms of self-compassion (*Yarnell et al., 2019*).

Of primary concern, the study design was cross-sectional hence it must be highlighted that longitudinal studies concerned with the development of self-compassion and fusion as well as its effects on decreasing self-stigma are required in order to provide an evaluation of causal relationships among the variables studied. This provides opportunities for future research in terms of acceptance and commitment therapy (or training) in parents of children with ASD focused on self-stigmatizing thoughts and social contexts of care in order to develop self-compassion and defusion. Previous research shows that micro-, two-session interventions (*Blackledge & Hayes, 2006*; *Hahs, Dixon & Paliliunas, 2019*) or short ACT training sessions (*Poddar, Sinha & Mukherjee, 2015*) are effective in terms of enhancing quality of life and decreasing negative symptoms in this group, as well as in reducing self-stigma in other minorities (*Yadavaia & Hayes, 2021*; *Luoma et al., 2008*). Perhaps development of other aspects of psychological flexibility–*e.g.*, self-as-context–would be of use.

## ACKNOWLEDGEMENTS

We would like to thank Ada Stasiak, Karolina Kawa, Oliwia Kuczka and Weronika Szubert from the University of Silesia in Katowice for their assistance in obtaining data.

### Funding

This publication was financed by the funds granted under the Research Excellence Initiative of the University of Silesia in Katowice. The funders had no role in study design, data collection and analysis, decision to publish, or preparation of the manuscript.

### Grant Disclosures

The following grant information was disclosed by the authors:
Research Excellence Initiative of the University of Silesia in Katowice.

### Competing Interests

The authors declare that they have no competing interests.

## Author Contributions

- Anna Pyszkowska conceived and designed the experiments, performed the experiments, analyzed the data, prepared figures and/or tables, authored or reviewed drafts of the paper, and approved the final draft.
- Krzysztof Rożnawski performed the experiments, authored or reviewed drafts of the paper, and approved the final draft.
- Zuzanna Farny performed the experiments, authored or reviewed drafts of the paper, and approved the final draft.

## Human Ethics

The following information was supplied relating to ethical approvals (*i.e.*, approving body and any reference numbers):

This study was approved by the Ethics Committee of the University of Silesia in Katowice (approval no.: KEUS.92/02.2021).

## Data Availability

The raw measurements are available in the Supplementary File.

## Supplemental Information

Supplemental information for this article can be found online at http://dx.doi.org/10.7717/peerj.12591#supplemental-information.

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
