# Peer review of "Self-stigma and cognitive fusion in parents of children with autism spectrum disorder. The moderating role of self-compassion"

_PeerJ, doi:10.7717/peerj.12591_

## Round 0.1 · original submission · Minor Revisions

First of all, let me apologize for the delay. I was waiting for a further reviewer, but s/he is so late that I decided to make a decision based on the two reviewers I received. As you will see, the opinions of the reviewer are quite convergent; both reviewers appreciate your work and find it convincing, and I agree with their assessment. The reviews are thoughtful and clear, so please refer to them for details.

I agree with Reviewer 1 that you should refer to mothers instead of parents, and add some considerations on the possibility to investigate the role of fathers in further research. I agree with Reviewer 2 that a grammar check is needed.

Looking forward to your revised contribution,

·

Basic reporting

• The article is written in English and use clear, unambiguous, technically correct text and it’s conformed to professional standards of courtesy and expression.

Please, check Line 151: include bracket before Carvalho

• Literature references provide sufficient field background/context.

Many DOI are not indicated: are you sure they are not available?

• The article includes good and comprehensive introduction and background to demonstrate how the work fits into the broader field of knowledge. Relevant prior literature is appropriately referenced.
• The structure, figures, tables fit the characteristics of a professional article and the figures are relevant to the content of the article, of good resolution, and appropriately described and labeled.
• Raw data are shared in accordance with our Data Sharing policy.
• The structure of the article conforms to an acceptable format of ‘standard sections.

• The submission is ‘self-contained’ and represents an appropriate ‘unit of publication’, The results are relevant to the hypothesis and they are not inappropriately subdivided merely to increase publication count.

Experimental design

• The submission clearly defines the research question, which is relevant and meaningful from an experimental and clinical point of view. The knowledge gap being investigated is identified (the relationship between self-stigma and cognitive fusion among parents of children with ASD), and statements are made as to how the study contributes to filling that gap.
• The investigation has been conducted rigorously and to a sufficient technical standard.
• The research has been conducted in conformity with the prevailing ethical standards in the field.
• Methods is described with sufficient information, but more information could improve the presentation of the tools in order to be reproducible by another investigator.
1) The Authors provide information material relating to the questionnaire authors' permission to perform the translation. It would be important to know if a back to back translation method has been followed or, if it has not been followed, to explain why it was not considered necessary.
2) Furthermore, since the questionnaire was proposed to a clinical population, it would be useful to specify whether a preliminary pilot testing was carried out with a Polish normative sample. Also in this case, if a preliminary pilot testing has not been done, it would be useful to specify why it was not considered necessary at the moment.
3) These suggestions are related to the hypothesis that the constructs explored (self stigma, fusion and self compassion) are undergoing cultural influences.

Validity of the findings

• The article is a replication study and – at the same time – a novelty both because it values the influence of self compassion and not mindfulness and because it applies psychological knowledge in the field of ASD research. Authors explain the rationale for the replication and describe clearly how it adds value to the literature.
• The data on which the conclusions are based, are provided in an acceptable discipline-specific repository. The data are robust, statistically sound, and controlled, but one limit could be considered during the presentation of the results and into the discussion:
Authors always speak of parents, but, actually, the sample is made up of 218 mothers out of 233 subjects. The validity of the study, therefore, concerns the mothers and not the parents. It could at least be useful include into the discussion the hypothesis that fathers may function better or worse than mothers and therefore may represent a moderating factor or a risk factor that accumulates to others. So, future research could values this hypothesis.

Please check the Line 229 - “while controlling for gender” -: it might be useful to specify whether this control makes sense given the extreme disparity in the number of females compared to males and I didn’t find this analysis in the results

Additional comments

I appreciated the study because it can provide support for psycho-educational interventions related to cognitive psychology to help mothers and fathers coping with the problem behaviors of children diagnosed with ASD. Work to implement self compassion instead of attention to symptoms of depression and anxiety can avoid the clinical designation of parents

Reviewer 2 ·

Basic reporting

Overall, the paper is well-written and the flow is logical and clear. There are a few minor grammatical errors. I have mentioned several of these in the review. But, the authors should run a grammar check after all the edits to ensure that the rest of the text is checked and edited prior to the submission of the revision. This paper addresses an important area and has interesting findings, It is recommended for publication after minor revision.

Experimental design

Research design and methods are clear.

Validity of the findings

The findings are valid and clear.

Additional comments

INTRODUCTION:

The introduction is well-written and describes the key concepts clearly for the reader. There are only a few minor points to address:

Line 134:
Please add ‘a’ between ‘with’ and ‘different’

Line 136:
Please and in ‘one’s’ and change ‘toward’ to ‘towards’. The edited version is as follows:
‘… to relate kindly to one’s own experiences and to be caring towards oneself’

Line 160:
The term ‘autism disorder’ is used here. It would be better to use the standardised term ‘autism spectrum disorder (ASD)’ at the beginning of the manuscript and subsequently just use ‘ASD’ throughout the text, for uniformity.

METHODS

Line 207:
Please add ‘the’ (‘… items adapted from the Self-compassion Scale …’)

Lines 201-202:
Please recheck the wording. It may be best to rephrase as “Most people blame parents for their children’s ASD” instead of: “Most people do blame parents for the ASD of their children”.

Lines 240-241:
For clarity, I suggest that the authors add the word ‘levels’ as follows: ‘various levels of ASD-related intellectual disability’. The original phrase implies that there are different types but not levels of intellectual disability.

DISCUSSION:

Line 319:
Please change ‘decrease’ to ‘decreasing’.

Line 361:
Please rephrase as follows: ‘… as well as be less likely to suppress …’.

Line 372:
Please replace ‘strong’ with ‘apparent’.

Line 381:
Rephrase as such: ‘… were less likely to experience negative …’.

Lines 382-383:
‘…low self-compassionate attitude towards oneself.’ can be rewritten as just ‘… low self-compassion.’

TABLE 1:
Please replace ‘jobless’ with unemployed. Were homemakers (housewives/househusbands) considered under the full-time occupation group?

FIGURES:
All the figures are clear and well-presented.

---

## Round 0.2 · accepted · Accept

I am happy to inform you that your manuscript has been accepted for publication.